# MicroRNAs in Small Extracellular Vesicles Indicate Successful Embryo Implantation during Early Pregnancy

**DOI:** 10.3390/cells9030645

**Published:** 2020-03-06

**Authors:** Qiang Tan, Shuang Shi, Jingjie Liang, Xiaowei Zhang, Dingren Cao, Zhengguang Wang

**Affiliations:** 1College of Animal Science, Zhejiang University, Hangzhou 310058, China; tq19952010@126.com (Q.T.); 15945192301@163.com (S.S.); 11617011@zju.edu.cn (J.L.); 11717013@zju.edu.cn (D.C.); 2Zhejiang Animal Husbandry Techniques Extension Station, Hangzhou 310020, China; 21717006@zju.edu.cn

**Keywords:** small extracellular vesicles, miRNAs, embryo implantation, recurrent implantation failure

## Abstract

Synchronous communication between the developing embryo and the receptive endometrium is crucial for embryo implantation. Thus, uterine receptivity evaluation is vital in managing recurrent implantation failure (RIF). The potential roles of small extracellular vesicle (sEV) miRNAs in pregnancy have been widely studied. However, the systematic study of sEVs derived from endometrium and its cargos during the implantation stage have not yet been reported. In this study, we isolated endometrium-derived sEVs from the mouse endometrium on D2 (pre-receptive phase), D4 (receptive phase), and D5 (implantation) of pregnancy. Herein, we reveal that multivesicular bodies (MVBs) in the endometrium increase in number during the window of implantation (WOI). Moreover, our findings indicate that CD63, a well-known sEV marker, is expressed in the luminal and glandular epithelium of mouse endometrium. The sEV miRNA expression profiles indicated that miR-34c-5p, miR-210, miR-369-5p, miR-30b, and miR-582-5p are enriched during WOI. Further, we integrated the RIF’s database analysis results and found out that miR-34c-5p regulates growth arrest specific 1 (GAS1) for normal embryo implantation. Notably, miR-34c-5p is downregulated during implantation but upregulated in sEVs. An implication of this is the possibility that sEVs miR-34c-5p could be used to evaluate uterine states. In conclusion, these findings suggest that the endometrium derived-sEV miRNAs are potential biomarkers in determining the appropriate period for embryo implantation. This study also has several important implications for future practice, including therapy of infertility.

## 1. Introduction

Infertility and subfertility are significant challenges in human reproduction. Despite great advancements in assisted reproduction technology (ART), particularly in embryo transfer technology improving fertility efficiency, implantation rates remain low. The challenge regards recurrent implantation failure (RIF) caused by inadequate uterine receptivity and insufficient communication between the embryo and the uterus [1,2]. Previous studies have revealed that pregnancy failure mainly occurs during the embryo implantation period [3]. In most cases, it is very difficult to master the balance between the developing embryo and maternal uterus. Embryo implantation, a complex but vital step in pregnancy establishment in mammals, occurs in a limited period. The process is regulated by multiple molecules consisting of microRNAs, cytokines, growth factors, and lipids [4]. Embryo implantation is established when the uterus reaches receptivity status and the embryos are competent to adhere [5]. Successful implantation requires adequate communication between the embryo and the maternal endometrium [6]. However, poor understanding of maternal–fetal crosstalk contributes to implantation rates.

miRNAs are small noncoding RNAs, 19–22 nucleotides long, that regulate the expression of endogenous genes by targeting the 3′ untranslated region (3′ UTR) to inhibit protein translation and mRNA degradation [7]. miRNA is one of the most important approaches to regulate gene expression during embryo implantation [8]. miRNAs, such as miR-200c [9] and miR-30d [10], took part in the implantation of blastocyst, and regulation of uterine receptivity had been verified [11,12]. Notably, miRNAs can be packaged into small extracellular vesicles (sEVs) and delivered to both the embryo and the endometrium to aid in fetal-maternal communication [13,14].

Exosomes, a subtype of extracellular vesicles, are coated with a lipid bilayer of 40–150 nm diameter. A significant number of sEVs were considered exosomes. sEVs are released by almost all types of cells [15]. Accumulating evidence suggests that the generation and release of sEVs is a crucial strategy in cellular communication [13]. The cargos of sEVs affect biological processes of recipient cells during physiological and pathological states [16]. There is growing evidence suggesting the potential roles of sEVs in early pregnancy [17,18,19]. sEVs are present in the viscous fluid secreted by the uterus throughout the menstrual cycle into the uterine cavity [17,20]. The uterine fluid (UF) secretion cargos vary with the remodeling of the endometrium architecture throughout the menstrual cycle [21,22]. Notably, sEVs act as biomarkers early diseases diagnosis [23,24] as well as embryo implantation [25]. Hence, sEVs are potential biomarkers evaluating the endometrial status at each specific phase, for instance, the window of implantation (WOI). Unfortunately, current studies could not provide a good understanding of sEVs for uterine physiological biopsy during early pregnancy.

In this study, we examined endometrium morphology during the preimplantation, implantation, and postimplantation periods in a mouse model. We isolated and identified endometrium-derived sEVs in these stages of early pregnancy. We analyzed the miRNAs profile in sEVs. These miRNAs were identified to be associated with successful implantation. The main aim of our study is to identify sEV miRNAs derived from the endometrium during early pregnancy that could be used as potential biomarkers for the evaluation of endometrial physiology. Moreover, the identified sEV miRNAs also could be crucial in monitoring infertility as well as for the development of clinic therapy and for infertility.

## 2. Materials and Methods

### 2.1. Cell Culture

Ishikawa and HEC-1-A cell lines were purchased from cell bank of Chinese Academy of Science (Shanghai, China). Ishikawa cells were cultured in dulbecco’s modified eagle medium (DMEM) culture medium (Gibco, Waltham, MA, USA) supplemented with 10% fetal bovine serum (FBS). HEC-1-A cell lines were cultured in McCOY’s 5A (Sigma, St. Louis, MO, USA) supplemented with 10% FBS. For primary endometrium cells (pECs) isolation, the mice on D4 of pregnancy were used. In brief, the uterus was removed into phosphate buffer saline (PBS) supplemented with 2% penicillin-streptomycin solution. Then, fat tissues and blood were carefully removed. Uterine tissues were cut into small pieces and digested with collagenase I. FBS was used to end digestion. The dispersed tissues were then filtered with a 100-μm cell strainer, and the supernatants were centrifuged at 1000 rpm for 5 min. pECs were cultured in DMEM culture medium supplemented with 10% FBS.

### 2.2. Small Extracellular Vesicles (sEVs) Isolation

Adult ICR (Institute of Cancer Research) female mice (8 weeks) were mated with fertile males randomly. The experimental mice were housed in a specific pathogen-free facility, and the experimental protocols were approved by the College of Animal Science, Zhejiang University Ethics Committee (No. 16485, 3/12/2019). We isolated sEVs from the mice uterus on D2, D4, and D5 of pregnancy. Briefly, mice were perfused through the heart at the left ventricle with cold PBS at a rate of 5 mL/2 min to remove blood from the tissues. The cervix was ligated carefully to avoid the loss of uterine fluid, and the excess fat and mucous tissues were removed from the uterus. The uterine horn was washed with 1 mL of cold PBS on each side. The fluid was then collected. Next, the endometrium was scraped from the uterus and digested with 0.25% Trypsin solution for 30 min. Exosome-free FBS was used to terminate the effect of Trypsin. Tissues’ debris was removed by ephemeral centrifugation. Then, the supernatants were mixed with the uterine fluid. The liquid was centrifuged at 600× *g* for 5 min to remove dead cells, at 1200× *g* for 15 min to remove the debris, and at 10,000× *g* for 30 min to remove microvesicles. The supernatants were filtered through a 0.22-μm Millipore. The clarified fluid was centrifuged using a type 70 Ti rotor (Beckman, Kraemer Boulevard Brea, CA, USA) at 120,000× *g* for 75 min twice. The pelleted sEVs were suspended in PBS for downstream experimental protocols.

### 2.3. Immunohistochemistry

The uteri were collected from mice on D2, D4, and D5 of pregnancy. The uteri were fixed in 4% paraformaldehyde, embedded in paraffin, and sectioned (5 μm). After deparaffinized, the sections were incubated in boiling 10 mM citrate buffer (pH 6.0) for antigen retrieval. Then, the sections were blocked with 10% FBS and incubated with antibody CD63 (Abcam, Cambridge, UK; Cat. No. ab217345) (1:100) overnight at 4 °C. Sections were washed in PBS and incubated with biotinylated secondary antibody for 2 h at room temperature. The image was captured by NIS Elements software using light microscopy (NIKON INSTRUMENTS, Shanghai, China).

### 2.4. Small Extracellular Vesicles (sEVs) Tracking Analysis

Small extracellular vesicles (sEVs) were incubated with1,1’-dioctadecyl-3,3,3’,3’-tetramethylindocarbocyanine perchlorate (DiI) (Beyotime, Shanghai, China; Cat. No.C1036) dye for 15 min at room temperature, and 5% Albumin from bovine serum (BSA) was used to avoid overstaining. The labeled sEVs were diluted with PBS and centrifugated at 120,000× *g* for 75 min. Finally, the pellets were resuspended in PBS. Then, the labeled sEVs were added into pEC cell cultures supplemented with 10% exosome-free FBS and incubated for 12 h. Cells were washed and fixed with 4% paraformaldehyde. Following, the cells were treated with 0.1% Triton X-100 for 30 min and incubated with a Fluorescein isothiocyanate isomer (FITC)-conjugated phalloidin (Solarbio, Beijing, China; Cat. No. CA1620). 2-(4-Amidinophenyl)-6-indolecarbamidine dihydrochloride (DAPI) (Beyotime, Shanghai, China; Cat. No.C1002) was used to mark nuclei. The cells were visualized by a confocal microscopy (ZEISS, Oberkochen, Germany) and captured or processed using ZEN 100 software. Exosomes-depleted FBS was prepared by ultracentrifugation overnight at 100,000× *g* at 4 °C.

### 2.5. Transmission Electron Microscopy (TEM)

For identification of sEV, sEVs were placed onto the formvar carbon-coated coper grids at room temperature (RT) for 1 min. The excess suspension was removed using a filter paper. Then, sEVs were rinsed in double distilled water thrice. Following, sEVs were stained using 2% uranyl acetate at RT for 1 min. The grids were then dried in air. Image was observed with a TecnaiG2 Spirit120KV transmission electron microscope operating at 120 kV (Thermo FEI, Rockford, IL, USA).

For observation of multivesicular bodies (MVBs) in endometrium, a standard TEM method was used. Briefly, fresh tissues were fixed with 2.5% glutaraldehyde overnight at 4 °C and washed with PBS. Tissues were fixed again with 1% osmic acid, stained using 2% uranyl acetate, dehydrated in gradient alcohol, and embedded in epoxy resin. Ultrathin sections were transferred onto carbon-coated coper grids and examined using a TecnaiG2 Spirit120KV transmission electron microscope operating at 120 kV (Thermo FEI).

### 2.6. Scanning Electron Microscopy (SEM)

For characterization of the morphological structures of luminal epithelium in early pregnancy, a SEM process was performed. In brief, endometrium tissues were fixed with 2.5% glutaraldehyde overnight at 4 °C and rinsed in water. Then, the tissues were fixed again with 1% osmic acid, stained with 2% uranyl acetate, and dehydrated in gradient alcohol. Following, the samples were dried out of liquid CO_2_ at critical point and coated with platinum. The tissues were scanned with a Nova Nano 450 scanning electron microscopy, and images were captured (Thermo FEI, Rockford, IL, USA).

### 2.7. Nanoparticle Tracking Analysis (NTA)

NTA was used to determine the sizes distribution and concentration of sEVs. sEVs were resuspended and diluted with PBS for analysis using ZetaView PMX 100 (Particle Metrix, Meerbusch, Germany). Particle movement was analyzed using NTA 8.02.28 software (ZetaView). For each group, at least three independent experiments were performed.

### 2.8. Western Blot Analysis

The protein concentration was measured by a BCA Protein Assay Kit (Beyotime; Shanghai, China; Cat. No. P0010S). Total proteins were dissolved using 6–12% SDS-PAGE and transferred to the polyvinylidene difluoride membranes (Millipore, Massachusetts, USA). The membranes were then blocked with QuickBlockTM western buffer (Beyotime; Cat. No. P0231) for 20 min at room temperature and incubated with primary antibody including CD63 (Abcam), CD9 (Diagbio; Hangzhou, China; Cat. No. db919), ALIX (Diagbio; Hangzhou, China; Cat. No. db3856), HSP70 (Diagbio; Hangzhou, China; Cat. No. db2396), Calnexin (Abclonal, Wuhan, China; Cat. No. A0803), GAPDH (glyceraldehyde-3-phosphate dehydrogenase) (Diagbio; Hangzhou, China; Cat. No. db106), and GAS1 (absin, Shanghai, China; Cat. No. abs141177) at 4 °C overnight. Following, the membranes were incubated with horseradish peroxidase-conjugated secondary antibodies (Abclonal, Wuhan, China; 1:3000) at room temperature for 2 h. The blots were detected using BeyoECL Plus (Beyotime; Shanghai, China; Cat. No. P0018S).

### 2.9. mRNA Datasets Analysis

The Gene Expression Omnibus (GEO) (https://www.ncbi.nlm.nih.gov/geo/) is a publish database for storing high-throughput gene expression datasets. Through the retrieval of the GEO database by keywords (*Homo sapiens*, endometrial receptivity, WOI, recurrent implantation failure, embryonic implantation, and miRNA), four gene expressed chips associated with RIF of humans (GEO Database: GSE111974, GSE103465, GSE26787, and GSE92324) were retrieved. The expressed data of microarrays were normalized before analysis. Limma package was employed in R basic to identify differentially expressed genes (DEGs), and |logFC| > 1.0 and *P* value < 0.05 were set as the thresholds. Then, the DEGs were screened, and drawing custom Venn diagrams (https://bioinformatics.psb.ugent.be/webtools/Veen/) was used to compare the DEGs in four datasets. Five miRNA–mRNA relation prediction databases (Targetscan, miRwalk, miRDIP, miRSearch, and miRtarBase) were applied to predict the target genes of miRNAs.

### 2.10. miRNA Agomir Injection

MiR-34c-5p agomir was purchased from the GenePharma Company (Shanghai, China). Agomir was diluted to the concentration 2 OD/5 μL. A surgical operation was performed for the mouse on D3 of pregnancy. FiveμL miR-34c-5p agomir was injected into one side of the uterine horn, and 5 μL DEPC (diethyl pyrocarbonate) water was injected into the other side of the uterine horn as a control. The number of implantation sites was checked on D7 of pregnancy.

### 2.11. RNA Extraction and qRT-PCR

Total RNAs were extracted from the mouse endometrium tissues using Trizol reagent (TIANGEN, Beijing, China; Cat. No. DP421), according to the manufacturer’s protocols. The antisense strand of RNAs was synthesized by the first strand cDNA systhesis kit (TIANGEN, Beijing, China; Cat. No. KR118). The relative expression of mRNA level was detected using SuperReal PreMix Color (SYBR Green) qRT-PCR kit (TIANGEN, Beijing, China; Cat. No. FP215). miRNAs were isolated from endometrial tissues, cells, and sEVs using miRNeasy Mini Kit (QIAGEN, Frankfurt, Germany; Cat. No. 217184). Then, RNA was eluted in RNase-free water and reverse-transcribed to cDNA following the kit protocol (TIANGEN, Beijing, China; Cat. No. KR211). cDNA samples were run using miRcute Plus miRNA qPCR Kit (SYBR Green) (TIANGEN, Beijing, China; Cat. No. FP411). GAPDH, U6, or cel-miR-39 were used as a control. Relative miRNA and mRNA expression levels were assessed by the 2^−△△Ct^ method. All samples were tested in triplicate at least. The information of primers as follows: 

GAPDH: forward, ACAACTTTGGTATCGTGGAAGG; reverse, GCCATCACGCCACAGTTTC. cel-miR-39: TCACCGGGTGTAAATCAGCTTG. U6: AACGAGAAGCGAACCAAAAAAA. Human GAS1: forward, ATGCCGCACCGTCATTGAG; reverse, TCATCGTAGTAGTCGTCCAGG. Mouse GAS1: forward, CCATCTGCGAATCGGTCAAAG; reverse, GCTCGTCGTCATATTCTTCGTC.

### 2.12. Statistical Analysis

All experiments in this study were performed three independent times at least. The values were reported as mean ± s.e.m (standard error of mean). Student’s t-test was employed to compare the significance in two groups. One-way analysis of variance (ANOVA) was performed to compare multiple (>2) means. Plots were used GraphPad Prism (GraphPad, San Diego, CA, USA) 6.0 or R basic 3.6.0 (University of Auckland, New Zealand). *P* value ≤ 0.05 was considered significant: ** *P* values ≤ 0.01, * *P* values ≤ 0.05, and ns *P* values ≥ 0.05.

## 3. Results

### 3.1. Extracellular Vesicles are Present and Secreted by the Endometrium

The cell-surface of the uterine epithelial cells undergoes drastic morphological changes during the peri-implantation stage. Notable changes occur in the microvilli on the apical membrane of the cells [26]. The SEM results revealed that the microvilli on the luminal epithelium shortened during the WOI (D4 and D5) compared to the preimplantation stage (D2) (Appendix A). Pinopode, flattened and smooth protrusions on the endometrium surface that always appear during WOI [27], were also observed on D4 mouse endometrium of the pregnancy (Appendix A). This implies that the uterus is in the period of implantation. Surprisingly, vesicles similar to sEVs were also observed on the surface of the endometrium (Appendix A; red arrows indicated). Current studies consider that the sEVs originate from the early endosome and package in the multivesicular bodies (MVBs) [28]. Additionally, during pregnancy, a significant number of molecular components, including sEVs, is released into the uterine cavity to regulate the physiological processes [6,29]. Hence, to gain further insights into the release of sEVs by the endometrium in early pregnancy, we performed TEM to investigate MVBs in the endometrium on D2, D4, and D5 of the pregnancy. The TEM results revealed that MVBs containing typical intraluminal vesicles (ILVs) present in the endometrium during the early stage of pregnancy (Figure 1a). These MVBs and ILVs had a distinct lipid bilayer, and the size of ILVs was similar to the sEVs. In the three different physiological stages of the endometrium during early pregnancy, no significant difference was identified on the size and number of MVBs (Figure 1c,d). However, the number of ILVs per MVB was significantly increased during WOI compared to the pre- and postimplantation stages (Figure 1b). An interesting phenomenon was also noticed: EVs were observed in the extracellular space (Appendix A). These data indicated that the endometrium secreted sEVs.

### 3.2. Characterization of the Endometrium Derived Small Extracellular Vesicles (sEVs)

To confirm the secretion of sEVs by the endometrium, we assayed for CD63, a well-documented marker of the sEVs [30]. CD63 is also crucial in embryo implantation [31]. The immunohistochemistry results indicate that CD63 was observed to be primarily localized in the luminal epithelium regions of the endometrium during early pregnancy (Figure 2). Notably, the CD63 was also observed in the glandular epithelium in WOI (D4) given that CD63 is enriched in sEVs, which had been shown to present in the endometrium during the early pregnancy. This implies that the endometrium secretes sEVs during the implantation stage. In this study, a mixture of UFand scraped/trypsin-treated endometrium were collected during the early stage of pregnancy. Using a standard ultracentrifugation protocol, we isolated sEVs from the endometrium (Figure 3a). We performed TEM to investigate the morphology of particles isolated from the endometrium. The results showed that these oval or bowl-shaped particles (Figure 3b) presented a typical morphology of what have been generally descripted for sEVs [15,16,32]. In addition, we found that more sEVs were observed in the field of vision on D4 of pregnancy. The western blot analysis showed enrichment of the sEV markers such as CD63, CD9, Alix, and HSP70 (Figure 3c). In the negative control, the endoplasmic reticulum protein Calnexin was absent in our sEV samples. To validate the purification of sEVs isolated using our methods, CD63, Alix, and Calnexin were also detected in the endometrium tissues and in the 500-g, the 1200-g, and the 10,000-g pellets. These results suggest that we purified the sEVs from the endometrium without any cell debris population (Figure 3c). Consistent with the TEM and the western blotting results, the NTA analysis indicated that the sizes of sEVs were enriched from 40 to 200 nm (Figure 3d). The mean sizes of the sEVs in each group were 133.9 nm, 137 nm, and 131.1 nm, respectively. On the other hand, particle number per ml of sEVs isolated from D4 uterus of pregnancy was significantly increased when compared to D2 and D5 (Figure 3e). These results demonstrated that the endometrium significantly secretes sEVs during the period of implantation. Endometrium-derived sEVs play a vital role in the regulation of the endometrial physiological environment. Next, we questioned whether sEVs are uptaken by the endometrial cells. Therefore, we separated the primary endometrium cells (pECs) and co-cultured with labeled sEVs derived from endometrium on D2, D4, and D5 of pregnancy. We observed a red fluorescence signal in the confocal images implying that the labeled sEVs were endocytosed by the pECs (Figure 3e). On the contrary, no fluorescence signal was observed in the negative control. Thus, our findings suggest that endometrium release sEVs into uterine lumen, which are delivered to the endometrial cells to regulate the uterine status during early pregnancy.

### 3.3. Endometrium-Derived sEVs Carry Receptive miRNAs

The establishment of uterine receptivity is a prerequisite for successful embryo implantation. MiRNAs have been reported to participate in the regulation of uterine receptivity and are present in sEVs mediating cell–cell communication. Dynamic changes in miRNA variations occur in early pregnancy, suggesting a vivid miRNA activity [11]. To investigate the presence of miRNAs in the sEVs released by endometrium, we analyzed the expression profile of selected 23 miRNAs in sEVs using qRT-PCR method. The selected miRNAs are involved in the regulation of uterus receptivity [10,11,12]. We found that the endometrium-derived sEVs on D2, D4, and D5 of pregnancy expressed all the selected miRNAs (Figure 4a). Among the identified miRNAs, most of the miRNAs were downregulated in the sEVs during the WOI and the postimplantation stage. However, the levels of miR-34c-5p, miR-210, miR-369-5p, miR-30b, and miR-582-5p were markedly increased in the sEVs during WOI compared to the preimplantation period. Most importantly, miR-34c-5p, miR-210, and miR-30b are confirmed to be packaged in sEVs and released by multi-types cells [33,34,35]. The expressed level of miR-34c-5p, miR-210, and miR-30b were significantly higher during implantation. In addition, in an attempt to reveal the potential function of the sEV-derived miRNAs up-/downregulated during implantation, we investigated that the mRNAs were potentially targeted by miRNAs in sEVs. The bioinformatics prediction analysis revealed that 2987 mRNAs were potential targets of these miRNAs, which were mainly enriched in the phosphatidylinositol 3 kinase(PI3K) /protein kinase B(AKT) signaling pathway, MAPK (mitogen-activated protein kinase) signaling pathway, focal adhesion, cell–cell adhesion, and extracellular exosome (Figure 4b–d).

### 3.4. miR-34c-5p Targets GAS1 (Growth Arrest Specific 1) in RIF (Recurrent Implantation Failure) Endometrium

Initially, we used four microarray RIF datasets (GEO Database: GSE111974, GSE103465, GSE26787, and GSE92324) in this study. The data were analyzed using R basic to screen for the DEGs between normal implanted and RIF endometrial tissues (with conditions: |logFC| > 1.0 and *P* value < 0.05). The DEGs of each dataset were plotted in a Venn diagram (Figure 5a). GAS1 (growth arrest specific 1) was determined to be the exclusive intersection gene. Detailed results of the top 400 DEGs in each chip are shown in Appendix A. The genes’ expression profile of the top 60 DEGs from GSE103465 and GSE111974 are shown in heat maps (Figure 5b,c, respectively). The expression of GAS1 was observed to be significantly lower in the RIF endometrium compared to the normal endometrial tissues of women (Figure 5d). Furthermore, our analysis predicted that the miRNAs regulate GAS1 through five miRNA–mRNA relation prediction databases (Targetscan, miRwalk, miRDIP, miRSearch, and miRtarBase). Detailed information of the predicted miRNAs (showing only the top 28 with the highest scores) in each database are shown in Appendix A. In our results, two miRNAs, has-miR-34c-5p and has-miR-34a-5p, were found by comparison of prediction results (Figure 5e). These results suggest that these miRNAs likely regulate the expression of GAS1 in the human endometrium. To further identify the regulated relationship between miR-34c-5p and GAS1, GSE71332 (a miRNAs expression chip of GEO database) was screened using bioinformatics approach. Only miR-34c-5p showed significantly lower expression in the normal fertile endometrial tissues (Figure 5f). The specific expression data of GAS1 and miR-34c-5p were depicted in Appendix A. Based on these results, we supposed that GAS1 had a positive function during embryo implantation. On the contrary, miR-34c-5p plays a negative role in this process, probably through regulating GAS1 expression. Hence, two human endometrial cell lines, Ishikawa and HEC-1-A, were used to validate the hypothesis. The Ishikawa cells present the feature for receptive endometrial cells as they are poorly polarized, serving as a good model for receptivity-associated research, while HEC-1-A cells display low adhesive properties for embryo adhesion, which makes them non-receptive endometrial cell lines [36]. Consequently, we found out that the expression level of GAS1 is significantly higher in the Ishikawa cells compared to HEC-1-A cells (Figure 5g). However, the expression level of miR-34c-5p was significantly higher in the HEC-1-A cells compared to the Ishikawa cells (Figure 5h). These results suggest that GAS1 was likely to involve the embryo implantation and is regulated by miR-34c-5p in the endometrium.

### 3.5. miR-34c-5p is Poorly Expressed during Embryo Implantation

The mature sequence of miR-34c-5p and its binding site in the 3′UTR of GAS1 were conserved both in human and mice (Figure 6a). In order to further validate the relationship between miR-34c-5p and GAS1 in the mouse model, the expressions of GAS1 and miR-34c-5p were examined in the endometrium during early pregnancy through the application of RT-qPCR (Figure 6b,c, respectively). In our results, the expression levels of GAS1 were significantly higher during WOI (Figure 6b). However, GAS1 was poorly expressed during the pre- and postimplantation periods, indicating the significant roles of GAS1 during embryo implantation. On the contrary, miR-34c-5p was significantly downregulated during WOI and upregulated pre- and postimplantation periods (Figure 6c). We also detected the protein level of GAS1 during early pregnancy. The results were consistent with the mRNA expression (Figure 6d). Next, we performed an injection experiment to further identify the regulation of GAS1 by miR-34c-5p in vivo. The expression level of miR-34c-5p was significantly lower in D2, D3, and D4; agomir of miR-34c-5p was injected into the uterus of pregnant mice on D2 of pregnancy. After 48 h of injection, the endometrium on D4 was collected to detect the expression level of GAS1 (Figure 6e). Interestingly, GAS1 was actually downregulated after injection of miR-34c-5p agnomir when compared to control group in the mouse model (Figure 6g). Thus, miR-34c-5p regulates GAS1 in vivo during embryo implantation. Further, we investigated the impact of miR-34c-5p on the number of implanted embryos. miR-34c-5p agomir was injected into uterus on D3, and the number of implanted embryos was examined four days later (Figure 6f). There was a significantly lower number of embryos inside of agomir-injected horns than on the other side of the control (Figure 6h). Therefore, we suggested that reinforcement of mmu-miR-34c-5p during WOI triggers a risk of embryo loss (Figure 6i). Collectively, these data confirmed a critical role of miR-34c-5p in maintaining normal embryo implantation during early pregnancy by targeting GAS1.

## 4. Discussion

In this study, sEVs were derived from a mixture of UFand scraped/trypsin-treated endometrium. The uterine secretions were complex and not well characterized, but the components in the uterine lumen of female mice or humans were likely derived primarily from the endometrium [25]. sEVs have been identified in UF during early pregnancy in a number of species, including humans and mice [17,20]. In addition, Endometrium epithelial cells cultures present sEVs. Hence, the main source of sEVs in extracellular environment of uterus is from the luminal/glandular epithelium [25]. According to the Minimal Information for Studies of Extracellular Vesicles (MISEV)2018 guideline [37], we classified the isolated vesicles in the 40–200-nm diameter range using “sEVs” instead of “exosomes”. In this study, the overall experiments were performed using the MISEV guidelines, including the characterization of sEVs regarding quantitation, morphology, size, and protein markers. Notably, there is increasing attention on using sEVs as a noninvasive diagnostic tool in various diseases [23,38].

Recently, there is considerable attention on the role of sEVs in signal communication between cells. The noncoding RNAs and proteins are packaged in sEVs and transferred into specific recipient cells where they perform their functions. As an important mediator of dialogue, an increasing interest of sEVs miRNAs have been centred [7]. Implantation involves intricate communication between embryos and the maternal endometrium, and sEVs are involved this process [29,39]. There were also direct evidences demonstrating that the release and uptake of EVs between the embryo and endometrial cells were bidirectional [40]. Embryo secretes miRNAs and sEVs miRNAs. MiRNAs derived from embryo contribute to the communication at the maternal-fetal interface and can be explored for embryo reproductive competence assessment [40]. The EVs derived from the endometrial cells or the inner cell mass of the blastocyst regulate the ability (migration and invasion) of embryonic trophoblast cells to establish successful implantation [41]. However, more research has been conducted on cell communication via extracellular vesicles in vitro, with less focus on the role of sEVs in the physiological status of uterus in vivo during implantation. Then, we conducted a comprehensive and sequential study of sEVs derived from the endometrium during implantation. We reported that the miRNAs in sEVs are associated with successful embryo implantation and provide a potential biomarker.

Changes in endometrial morphology are related to normal uterine physiology during the menstrual cycle in women. In early pregnancy, the architectural changes in uterus provide an environment for embryo implantation including epithelial–mesenchymal transition [42] and decidualization [43]. Here, we aimed to identify the physiological states of implantation. The SEM results indicated that the uteri collected on D2, D4, and D5 of pregnancy correspond to the preimplantation, implantation, and postimplantation periods, respectively. Studies have demonstrated that uteri secreted abundant factors into the uterine cavity to nurture embryos during early pregnancy [44]. In this study, we observed vesicles similar to sEVs located in the endometrial surface. Most conducted studies on the sEVs released by the endometrium use endometrial cell lines [18,20,25,45], with no in vivo studies on the sEVs derived from the endometrium during early pregnancy. sEVs primarily originate from the MVBs and ILVs and are released when the MVBs fuse with the plasma membrane [46]. The results of this study showed that MVBs are present in endometrium during implantation, and the number of ILVs per MVB significantly increases during WOI. In addition, our findings indicated that CD63 is mainly located in the uterine luminal and glandular epithelium. These results suggest that the endometrium secretes sEVs during early pregnancy in mice models.

In previous studies, exosomes/EVs have been isolated from the UF [17] but lack the study containing whole stages of early pregnancy from early to late secretory. We found out that sEVs were present in the UF and endometrial tissues. In view of this, we collected UF and endometrial tissues from the period of preimplantation, WOI, and postimplantation. sEVs were isolated and purified. Our findings revealed that sEVs derived from the endometrium contributed to the regulation of appropriate uterine physiological states for successful implantation. The organ- and tissue-secreted sEVs carried specific cargos in the different physiological stages [47]. sEVs fuse with the plasma membrane of recipient cells, and the contents of sEVs were transferred into the target cells. In our results, these sEVs could be delivered to the endometrial cells. In addition, sEV miRNAs were the main source of circulating miRNAs, and sEVs enriched miRNAs selectively with the development of diseases [48]. For instance, increased miRNAs in sEVs are associated with cancer development [49]. A previous study of human miRNA profiling revealed that maternal miRNAs were released into the UF during the WOI and were packaged in exosomes [20]. Unlike previous studies, we directly analyzed the expression profiles of miRNAs in sEVs rather than in the UF to avoid interference of other components in the UF. These miRNAs, which associated with uterine receptivity and regulated cell migration, cell invasion, cell adherence, and proliferation, mainly existed in UF. Hence, we propose a new method of determining the physiological status of the uterus. Infertility is a major issue in human reproduction, and in vitro fertilization is costly with a relatively low success rate, in general below 30% [3]. The inadequacy of uterine receptivity significantly contributes to failure during in vitro fertilization and embryo transplantation [50]. Increasing evidence indicates that sEVs miRNAs are potential biomarkers for the diagnosis and treatment of diseases [23,51,52]. In our study, miR-34c-5p, miR-210, and miR-30b were upregulated in sEVs during implantation. This suggests that these sEV miRNAs signal a suitable uterine environment for embryo implantation.

Notably, we aimed at identifying biomarkers for uterine physiological status indication. Embryo transplantation occurs at a suitable period when the uterus is receptive. Therefore, the uterine status and environment in RIF patients can be therapeutically stimulated for successful embryo implantation. Despite new technologies (artificial assisted reproduction) making major advancements towards improving fertility efficiency, RIF still remains a major issue in human reproduction health. Our results from that database analysis revealed that GAS1 associated with RIF. GAS1 plays a major role in embryonic development and human diseases and involved stem cell renewal and cancer growth [53]. However, the roles of GAS1 in embryo implantation have not been studied. Here, we report that GAS1 was upregulated in the receptive endometrium and downregulated in the non-receptive endometrium during early pregnancy, indicating the function of GAS1 during this process. Additionally, we found out that miR-34c-5p targeted GAS1 in the endometrium during embryo implantation. The suppressing effect of miR-34c-5p is necessary for embryo implantation during WOI. A series of studies have reported that miR-34c-5p involved the process of endometrial receptivity and inflammation [12,54]. An interesting study revealed that miR-34c-5p inhibited exosome release by targeting RAB27B, a molecule that promotes exosome shedding [34]. miR-34c-5p was downregulated during WOI; following, the secretion of sEVs was increased, possibly because of the regulation between miR-34c-5p and RAB27B. On the other hand, miR-34c-5p was found to be upregulated in the sEVs isolated from UF during WOI. This suggests that the miR-34c-5p could be a castoff of the endometrium, but the sEVs miR-34c-5p could potentially reflect the receptivity of the uterus. These findings suggest that the endometrium-derived sEVs miRNAs are potential biomarkers in determining the appropriate period for embryo implantation as aid in the development of therapeutic agents for RIF.

In summary, our findings reveal that sEVs are released by the endometrium in mice models during early pregnancy, and the miRNA profile of the secreted sEVs varies with the physiological states of the uterus. Moreover, miR-34c-5p plays a crucial role in embryo implantation. Finally, we identified miR-34c-5p in sEVs derived from the endometrium during implantation, which are potential receptive biomarker signaling of endometrium for successful embryo implantation.

## Figures and Tables

**Figure 1 cells-09-00645-f001:**
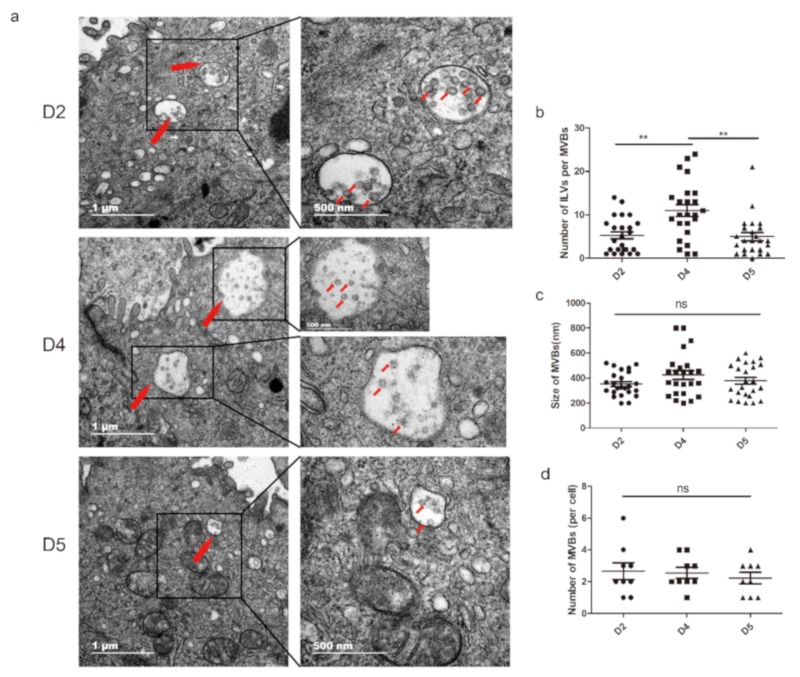
MVBs are present in mouse endometrium cells during early pregnancy. (**a**) Representative electron microscopic images of endometrium tissues on D2, D4, and D5 of pregnancy: Red arrows indicate MVBs including classic intraluminal vesicles (ILVs). (**b**) The number of ILVs per MVB, (**c**) size of MVBs, (**d**) number of MVBs per cell in different stages: MVBs were counted only when containing typical ILVs. ** *P* < 0.01. ns: not significant.

**Figure 2 cells-09-00645-f002:**
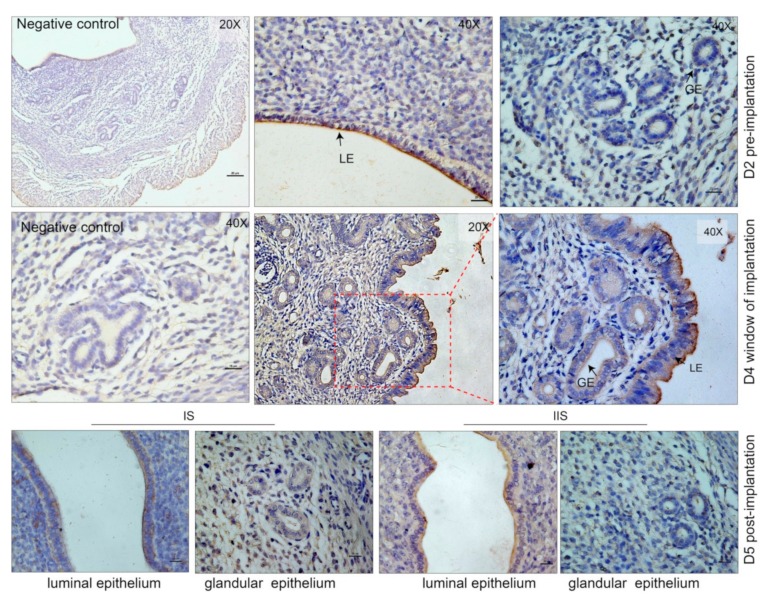
Immunohistochemical staining of CD63 in mouse glandular and luminal epithelium during pre-implantation (D2), window of implantation (WOI) (D4), post-implantation (D5) stage, and in the negative control (no primary antibody). LE: luminal epithelium. GE: glandular epithelium. IS: implantation site. IIS: inter implantation site.

**Figure 3 cells-09-00645-f003:**
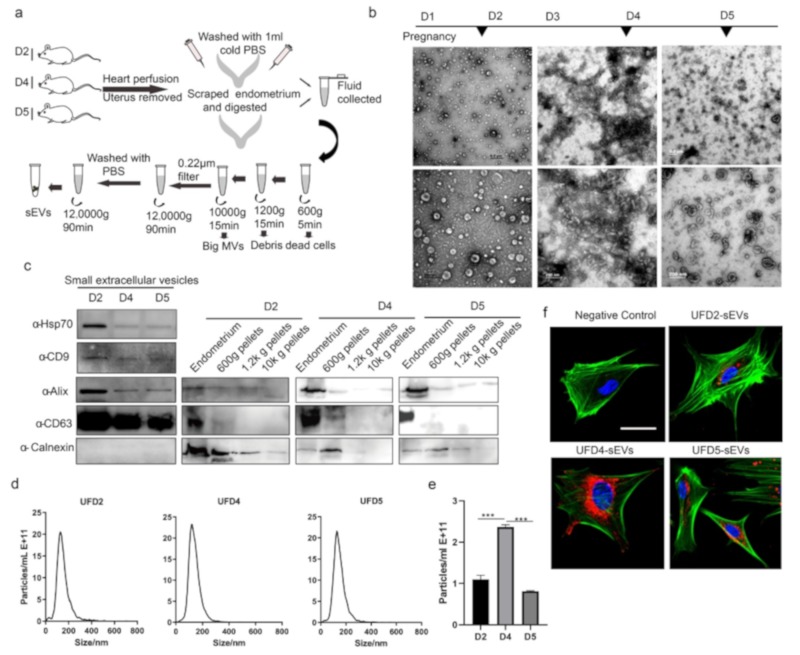
Identification of small extracellular vesicles (sEVs) derived from endometrium: (**a**) Schematic diagram showed the strategy of isolation and purification of sEVs derived from mouse endometrium during early pregnancy. (**b**) TEM images showed sEVs derived from mouse endometrium on D2, D4, and D5 of pregnancy. Magnification: top, 18,500. bottom, 68,000. (**c**) Western blotting analysis of sEV protein markers (CD63, Alix, CD9, and HSP70): Calnexin was used as a negative control. (**d**) Nanoparticle Tracking Analysis (NTA)-suggested size distribution and concentration of sEVs. (**e**) Concentration of sEVs derived from the uterus on D2, D4, and D5 of pregnancy. (**f**) Confocal microscopic images showed uptake of labeled sEVs by primary endometrium cells (pECs). Nuclei are stained by DAPI in blue, sEVs are stained by DiI in red, and actin is stained by FITC-conjugated phalloidin in green. (Bar 20 μm). *** *P* < 0.001.

**Figure 4 cells-09-00645-f004:**
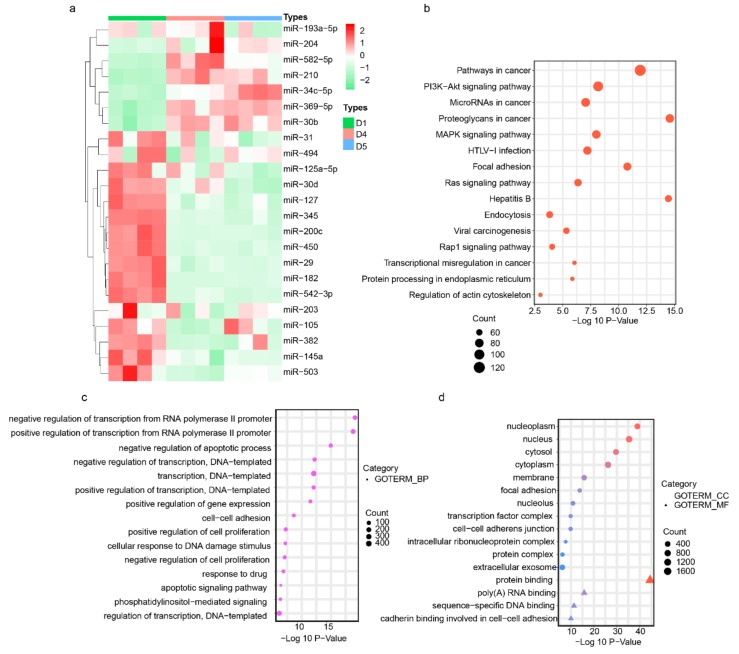
Endometrium-derived sEVs carry miRNAs which associated with uterine receptivity: (**a**) Heat map showed the expression profiles of miRNAs in sEVs derived from endometrium on D2, D4, and D5 of pregnancy. (**b**–**d**) GeneOntology GeneOntology (GO) and Kyoto Encyclopedia of Genes and Genomes (KEGG)terms of genes were potentially targeted by miRNAs in sEVs. (**b**) A bubble plot of KEGG pathways enriched. (**c**) A bubble plot of GOs (Biological Process) enriched. (**d**) A bubble plot of GOs (molecular function and cellular components) enriched.

**Figure 5 cells-09-00645-f005:**
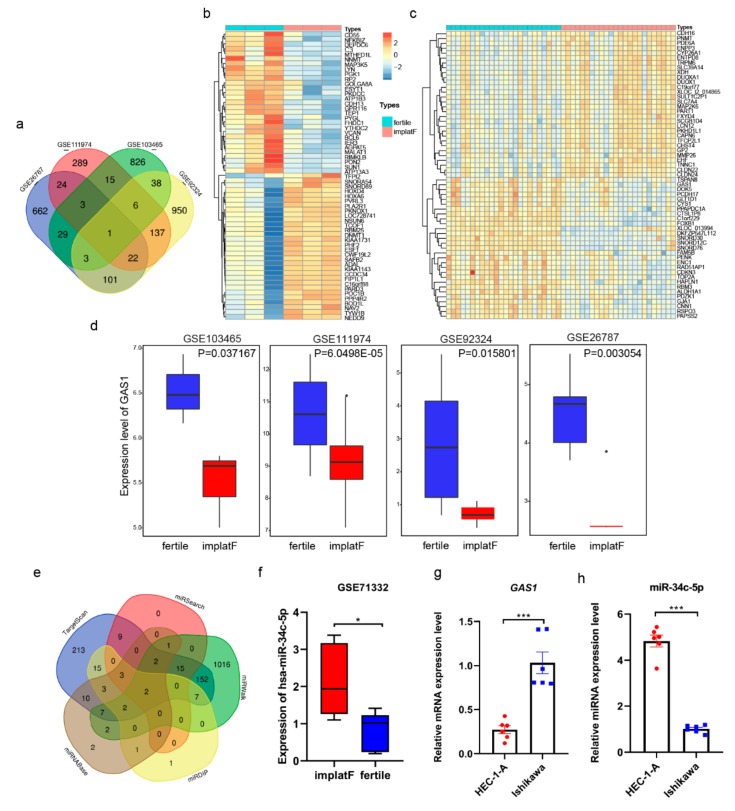
Growth arrest specific 1 (GAS1) was targeted by miR-34c-5p. (**a**) The Venn diagram of co-expressed differentially expressed genes (DEGs) in four Gene Expression Omnibus (GEO) databases (GSE26787, GSE111974, GSE103465, and GSE92324): only one gene, GAS1, was observed. (**b**,**c**) Heat maps show the top 60 differentially expressed genes in GSE103465 and GSE111974, respectively. (**d**) The expression of GAS1 in four gene microarrays. (**e**) The comparison of target miRNA of GAS1 predicted by TargetScan, miRSearch, miRTarBase, miRWalk, and mirDIP. (**f**) The expression of hsa-miR-34c-5p in miRNA expression chip GEO: GSE71332 of RIF. (**g**) The mRNA level of GAS1 in Ishikawa and HEC-1-A cell lines. (**h**) The expression of miR-34c-5p in Ishikawa and HEC-1-A cell lines. * *P* < 0.05, *** *P* < 0.001.

**Figure 6 cells-09-00645-f006:**
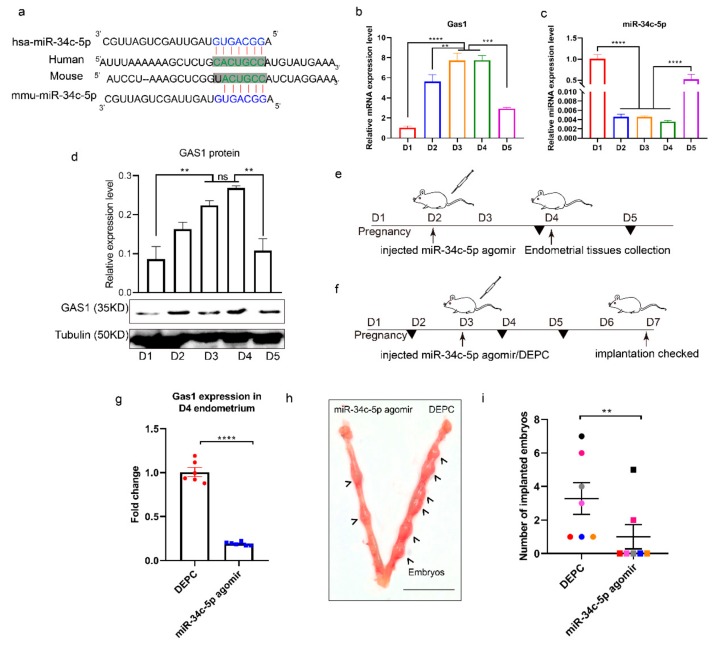
miR-34c-5p regulated embryo implantation by targeting GAS1 in endometrium during early pregnancy. (**a**) The regulating relationship between GAS1 and miR-34c-5p, the mature sequence of miR-34c-5p in mouse and human, was predicted by Targetscan. (**b**,**c**) The relative expression of GAS1 and miR-34c-5p in mouse endometrium was detected by qRT-PCR during early pregnancy. (**d**) The protein expression level of GAS1 in mouse endometrium. (**e**) miR-34c-5p regulated the expression of GAS1 in endometrium. (**f**) miR-34c-5p regulated successful embryo implantation. (**g**) The expression of GAS1 in D4 mouse endometrium after injecting miR-34c-5p agomir in D2. (**h**,**i**) The effect of miR-34c-5p agomir injected in vivo on the number of embryos. Bar = 1 cm. ** *P* < 0.01, *** *P* < 0.001.

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
