# Peer review of "MicroRNAs in Small Extracellular Vesicles Indicate Successful Embryo Implantation during Early Pregnancy"

_cells, 2020, doi:10.3390/cells9030645_

Round 1
Reviewer 1 Report
In this manuscript, Tan et al., characterize sEVs in the mouse endometrium using a range of microscopic and immune techniques. They then screen the sEVs for miRNA and narrow down mi-R-34c-5p as an important regulator of GAS1. The authors screened publically available datasets and demonstrate GAS1 is lower in patients with RIF. Murine in vivo experiments demonstrate mi-R-34c-5p inhibits embryo implantation. The investigation is sound and of general interest in reproductive biology. There is a developing field in role of miRNA in reproductive biology, especially as a means of dialogue between embryo and decidua, and this manuscript adds to this understanding. I would consider publication of a revised manuscript if the following points and extra experiments are addressed.
Major
- Could the authors please thoroughly proofread the manuscript? There were several typographical, structural, consistency, spacing and phrasing issues which made the manuscript hard to read. A few typos are highlighted below, but I have not listed everything because there is too much that needs changing. A lot of proofreading and editing is required to get the manuscript up to publication quality.
- Comparing figure 3c, with 1b and 2, the authors suggest CD63 is highest in the implantation window (day 4), yet in western blotting appears highest at day 2. Is this a western loading issue? The authors need to provide evidence of equal protein loading in western blots.
- I am not convinced by figure 2. The authors state that CD63 was expressed in the luminal and glandular epithelium. Staining is barely detectable by eye in glands, so this statement needs to be softened. Did the authors optimise antibody concentrations? It would be informative for the authors to quantify staining (image analysis software such as ImageJ has colour deconvolution add-ons which can separate brown (CD63) and haematoxylin). We could then compare across phases and have an idea of n-numbers (data that is missing from figure legend). This would justify the statement ‘increased significantly’ in line 227.
- Can the authors be sure that LE staining is specific? And not a result of poor washing techniques and coagulation of detection antibody on the outmost edge of tissue? (so called ‘edge effect’). The authors need to show their negative controls, where the CD63 primary antibody was excluded.
- The authors use DiI to show uptake of isolated sEVs in pECs. If this is the case, and as they suggest sEVs carry mi-R-34c-5p, we would expect a down regulation of GAS1 in these cells. The authors need to perform an uptake experiment and then assess the expression of GAS1 using PCR and/or western blotting. This would link the sEV characterisation to the rest of the manuscript and add weight to their study.
- For confirmation, the authors then need to repeat this experiment with inhibition of miRNAs. There are various chemical inhibitors of miRNAs available.
- Please find a way to quantify and show average data for figure 3e
- The authors have shown release and uptake of sEV. Could the authors elaborate in the discussion on which potential cells are releasing and up-taking sEVs. Do they act as autocrine or paracrine signals? I feel further details on the communication networks is needed.
- The authors discuss embryo derived signals, is there any evidence that embryos secrete miRNAs? Is there detailed evidence that they secrete mi-R-34c-5p? What effect could this have on receptivity? Or as a means of embryo quality detection? Could the authors comment in the discussion on the embryo-decidua communication.
Minor
- Section 3.1. The authors have used pinodes as a receptivity marker. The authors need to downplay this statement as there is controversy and considerable evidence (org/10.1093/humupd/dmn052 and 10.1023/a:1015455709833 that they do not represent receptivity.
- Figure 3a is confusing. You have 2x adjacent 12000g centrifuge steps. Was there washing between spins?
- Line 270. Please elaborate on methods as to how this data set was obtained? Were these screened from the sEVs isolated as per fig 3a? The manuscript isn’t clear. Also, please give the scale bar a unit in 4a. Is this a Z-score? Clarify throughout.
- Can the authors specify which species the data sets are from in the manuscript. Some clarity is needed.
- Line 16. Have ‘been’ repeatedly
- Line 21. Please briefly explain what CD63 is in abstract
- Line 78. Please explain what HEC-1 cells are
- Line 95. Change Exosme to ‘Exosome’
- Line 96. Change ‘were’ to ‘was’
- Line 155. Change ‘publish’ to ‘published’
- Line 169 and 170. Change ‘u’ to ‘µ’
- Line 177. Change ‘form’ to ‘from’
- Line 226. ‘epithelium’ repeated
- Figure 3. Change ‘Hesrt’ to ‘Heart’
- Line 258. Change ‘up’ to ‘top’, and ’down’ to ‘bottom’
- Figure 3c. Ensure western blots in fig 3c are easily identifiable. Keep blots the same size and in alignment with labels on the left
- Line 294. Change ‘showed’ to ‘shown’
- Line 332. Change ‘has’ to ‘have’
- Line 336. Change ‘poorly’ to ‘poor’
- Line 338. Change ‘detected’ to ‘detect’
- Line 342. Change ‘in’ to ‘on’
- Line 407. ‘were were’ please delete a ‘were’
- Line 405. Change ‘reveled’ to ‘revealed’
Author Response
Dear Reviewer:
Thank you for your comments concerning our manuscript. Those comments are all valuable and very helpful for revising and improving our paper, as well as the important guiding significance to our researches. We have studied comments carefully and have made correction which we hope meet with approval.
Responses to the reviewers' comments as follows:
All responses were based on the revised manuscript.
In this manuscript, Tan et al., characterize sEVs in the mouse endometrium using a range of microscopic and immune techniques. They then screen the sEVs for miRNA and narrow down mi-R-34c-5p as an important regulator of GAS1. The authors screened publically available datasets and demonstrate GAS1 is lower in patients with RIF. Murine in vivo experiments demonstrate mi-R-34c-5p inhibits embryo implantation. The investigation is sound and of general interest in reproductive biology. There is a developing field in role of miRNA in reproductive biology, especially as a means of dialogue between embryo and decidua, and this manuscript adds to this understanding. I would consider publication of a revised manuscript if the following points and extra experiments are addressed.
Major
Point 1: Could the authors please thoroughly proofread the manuscript? There were several typographical, structural, consistency, spacing and phrasing issues which made the manuscript hard to read. A few typos are highlighted below, but I have not listed everything because there is too much that needs changing. A lot of proofreading and editing is required to get the manuscript up to publication quality.
Response 1: Thank you for your suggestion about the quality of the writing. To further improve the writing quality, we turned to a professional English editing service to modify the sentences, words, and grammar in the paper to ensure its accuracy.
Point 2: Comparing figure 3c, with 1b and 2, the authors suggest CD63 is highest in the implantation window (day 4), yet in western blotting appears highest at day 2. Is this a western loading issue? The authors need to provide evidence of equal protein loading in western blots.
Response 2: Thank you for your suggestion. According to the MISEV2018 guideline (https://www.ncbi.nlm.nih.gov/pubmed/30637094), in order to identify sEVs, NTA analysis (figure 3d), TEM (figure 3b) and protein marker detection (figure 3c) must all be required. The purpose of figure 3c is to detect the protein marker of sEVs. Before western blot, the proteins concentration was measured by a BCA protein kit, equal protein loading was added into each well. The results of WB could not represent the concentration of sEVs. The purpose of figure2 is to detect and locate CD63 protein in endometrium.
Thus, we used NTA analysis to examine the concentration of sEVs derived from uterus during early pregnancy. As shown in new figure 3e of revised manuscript, the concentration of sEVs was significantly high in D4 of pregnancy. This result is consistent with figure1b and figure2.
Point 3: I am not convinced by figure 2. The authors state that CD63 was expressed in the luminal and glandular epithelium. Staining is barely detectable by eye in glands, so this statement needs to be softened. Did the authors optimise antibody concentrations? It would be informative for the authors to quantify staining (image analysis software such as ImageJ has colour deconvolution add-ons which can separate brown (CD63) and haematoxylin). We could then compare across phases and have an idea of n-numbers (data that is missing from figure legend). This would justify the statement ‘increased significantly’ in line 227.
Response 3: Thank you for your suggestion. We have revised the inappropriate interpretation and description of results and conclusions. I think the main reason for the unclear staining in Figure 2 is the brightness and resolution of the picture. In addition, our results are consistent with previous studies in human uterus, where CD63 is mainly localized in the luminal and glandular epithelium. The expression of CD63 was increased during implantation.
Point 4: Can the authors be sure that LE staining is specific? And not a result of poor washing techniques and coagulation of detection antibody on the outmost edge of tissue? (so called ‘edge effect’). The authors need to show their negative controls, where the CD63 primary antibody was excluded.
Response 4: Thank you for your suggestion. We have shown the negative immunohistochemical control in Figure 2. That is, CD63 primary antibody was excluded. In addition, the antibodies we use are monoclonal antibodies, so the staining is specific.
Point 5: The authors use DiI to show uptake of isolated sEVs in pECs. If this is the case, and as they suggest sEVs carry mi-R-34c-5p, we would expect a down regulation of GAS1 in these cells. The authors need to perform an uptake experiment and then assess the expression of GAS1 using PCR and/or western blotting. This would link the sEV characterisation to the rest of the manuscript and add weight to their study.
Point 6: For confirmation, the authors then need to repeat this experiment with inhibition of miRNAs. There are various chemical inhibitors of miRNAs available.
Response5 and 6: Thank you very much for your constructive suggestions, which are important for us to review our research. Your suggestion gives us a new direction. But this experiment is not necessary in the present study. The main purpose of our study was to screen for biomarkers. The regulation of GAS1 by miR-34c-5p may occur in cells. Because miR-34c-5p is almost not expressed in endometrial tissue during implantation, and miR-34c-5p is mainly enriched in sEVs, indicating that sEVs miR-34c-5p not regulates GAS1 expression.
Previous studies have shown that uterine-derived exosomes can be delivered to embryos, thereby promoting embryo implantation. This is an aspect funtion of uterine-derived exosomes. In addition, studies reported that exosomes secreted by cells act as paracrine effects that affect the function of surrounding cells, such as cancer cells. Then we speculate whether a large number of sEVs in the uterus also act as paracrine to regulate the uterine environment. So we did this uptake experiment, and the specific molecular mechanism will be studied in depth in our next work.
Point 7: Please find a way to quantify and show average data for figure 3e
Response7: Thank you for your suggestion. The purpose of Figure 3e is to demonstrate that endometrial cells are able to take up their own secreted sEVs. This is a qualitative experiment and there is no quantitative standard in most studies of sEVs.
Point 8: The authors have shown release and uptake of sEV. Could the authors elaborate in the discussion on which potential cells are releasing and up-taking sEVs. Do they act as autocrine or paracrine signals? I feel further details on the communication networks is needed.
Response8: Thank you for your suggestion. We have discussed the issues that you mentioned. There are two main types of cells in endometrium tissues, including stromal cells and epithelial cells. Extracellular vesicles have been identified in uterine fluid during the estrous/menstrual cycles in a number of species, including humans, sheep, and mice [18–20, 22]. Their release from endometrial epithelial cells in culture [18, 121–123] indicates the luminal/glandular epithelium as their source. Those sEVs influence the potency of embryo for implantation, while some of them act as autocrine or paracrine signals to regulate the uterine environment.
Point 9: The authors discuss embryo derived signals, is there any evidence that embryos secrete miRNAs? Is there detailed evidence that they secrete mi-R-34c-5p? What effect could this have on receptivity? Or as a means of embryo quality detection? Could the authors comment in the discussion on the embryo-decidua communication.
Response9: Thank you for your suggestion. According to your suggestion, we commented the embryo-decidua communication in the “discussion” section. Previous study had reported that embryo secrete miRNAs and sEVs miRNAs, but no evidence indicated that they secret miR-34c-5p. The miRNAs derived from embryo contribute to the communication at the maternal-fetal interface, and can be explored for embryo reproductive competence assessment.
Minor
Point 10: Section 3.1. The authors have used pinodes as a receptivity marker. The authors need to downplay this statement as there is controversy and considerable evidence (org/10.1093/humupd/dmn052 and 10.1023/a:1015455709833 that they do not represent receptivity.
Response10: Thank you for your suggestion. We have revised our statement in the manuscript. We observe the morphological characteristics of the endometrial surface through scanning electron microscopy to determine the state of the uterus in the early stages of pregnancy.
Point 11: Figure 3a is confusing. You have 2x adjacent 12000g centrifuge steps. Was there washing between spins?
Response11: Thank you for your suggestion. Yes,there has washing between spins. We have revised this diagram in figure 3a. sEVs were washed twice with PBS in 12000g centrifuge steps.
Point 12: Line 270. Please elaborate on methods as to how this data set was obtained? Were these screened from the sEVs isolated as per fig 3a? The manuscript isn’t clear. Also, please give the scale bar a unit in 4a. Is this a Z-score? Clarify throughout.
Response12: Thank you for your suggestion. We have elaborated on the acquisition of these data in the method section.
The data shown in the heat map in Figure 4a is the expression of miRNAs in sEVs. We extracted RNA from isolated sEVs (figure 3a) and performed qRT-PCR detection on miRNAs. We analyzed their relative expression using the 2-△△Ct method.
Point 13: Can the authors specify which species the data sets are from in the manuscript. Some clarity is needed.
Response13: Thank you for your suggestion. We have added the species information of data sets in “materials and methods--- mRNA datasets analysis” section.
Point 14:
Line 16. Have ‘been’ repeatedly
Response: Thank you for your suggestion. We have corrected this word.
Line 21. Please briefly explain what CD63 is in abstract
Response: Thank you for your suggestion. We have added the explanation of CD63 in abstract of revised manuscript.
Line 78. Please explain what HEC-1 cells are
Response: This is a spelling mistake. We have corrected “HEC-1 cells” to “HEC-1-A cells”.
Line 95. Change Exosme to ‘Exosome’
Response: Thank you for your suggestion. We have corrected “Exosme” to “Exosome”.
Line 96. Change ‘were’ to ‘was’
Response: Thank you for your suggestion. We have corrected “were” to “was”.
Line 155. Change ‘publish’ to ‘published’
Response: Thank you for your suggestion. We have corrected “publish” to “published”.
Line 169 and 170. Change ‘u’ to ‘µ’
Response: Thank you for your suggestion. We have corrected “u” to “µ”.
Line 177. Change ‘form’ to ‘from’
Response: Thank you for your suggestion. We have corrected “form” to “from”.
Line 226. ‘epithelium’ repeated
Response: Thank you for your suggestion. We have deleted repeated word.
Figure 3. Change ‘Hesrt’ to ‘Heart’
Response: Thank you for your suggestion. We have corrected “Hesrt” to “Heart” in new figure3 in revised manuscript.
Line 258. Change ‘up’ to ‘top’, and ’down’ to ‘bottom’
Response: Thank you for your suggestion. We have corrected these words.
Figure 3c. Ensure western blots in fig 3c are easily identifiable. Keep blots the same size and in alignment with labels on the left
Response: Thank you for your suggestion. We have adjusted the blots size in figure 3c and alignment with labels on the left.
Line 294. Change ‘showed’ to ‘shown’
Response: Thank you for your suggestion. We have corrected “showed” to “shown” in new figure3 in revised manuscript.
Line 332. Change ‘has’ to ‘have’
Response: Thank you for your suggestion. We have corrected “has” to “have”.
Line 336. Change ‘poorly’ to ‘poor’
Response: Thank you for your suggestion. We have corrected “poorly” to “poor”.
Line 338. Change ‘detected’ to ‘detect’
Response: Thank you for your suggestion. We have corrected “detected” to “detect”.
Line 342. Change ‘in’ to ‘on’
Response: Thank you for your suggestion. We have corrected “in” to “on”.
Line 407. ‘were were’ please delete a ‘were’
Response: Thank you for your suggestion. We have deleted a were’.
Line 405. Change ‘reveled’ to ‘revealed’
Response: Thank you for your suggestion. We have re-wrote this sentence.

Reviewer 2 Report
Tan and colleagues have investigated the endometrium and it products in pregnant mice at pre-implantation, implantation, and post-implantation phase. The number of intraluminal vesicles in the endometrium is increased during implantation. They obtained extracellular vesicles (EVs) from the endometrium and showed that these are full of microRNAs, including many that previously have been related to uterine receptivity. Notably, levels of miR-34c-5p in the EVs are increased during the window of implantation (d.4). Analysis of information from miR prediction databases and previous research of miR/mRNA expression of endometrial tissue (normal and RIF) showed that: 1) miR-34c-5p negatively influences GAS1 expression; and 2) implantation failure in humans is related to decreased GAS1 expression and increased miR-34c-5p expression. In their own mouse model the investigators demonstrate increased GAS1 and decreased miR-34c-5p during the window of implantation. Finally, they show that injection of miR-34c-5p agomirs into the uterine horns at d.3 of pregnancy leads to decreased GAS1 expression and a reduced number of implanted embryo (at d.7), confirming that indeed this gene plays a role in embryo implantation.
This is a nice study, incorporating both various practical techniques as well as database mining and mechanistic studies, to verify the hypothesis that microRNAs (in this case 34c-5p) is essential in determining success or failure of embryo implantation.
I do have doubts about the title and some of the main conclusions: what is now the evidence that it is the EVs that indicate successful embryo implantation? Previous research, on which data mining was performed by the authors, has been derived from endometrial tissues and not from vesicles. Furthermore, in their own mouse model (Fig. 6), the miR-34c-5p expression was determined in endometrial tissues. In other words, although Fig. 4 shows the expression profiles of miRs in EVs, it is not clear how much of the tissue (cytoplasm) itself contributes to the miR pool and how much is contributed by the EVs. I understand that there is need for biomarkers, possibly miR-containing EVs, of implantation failure, but currently the formulation of the title and some of the statements / conclusions throughout the manuscript should be adjusted (e.g. to something like “miR-34c-5p is enriched in EVs and decreases the success of embryo implantation”).
In the light of the previous comment, Fig. 3e is not completely clear to me: the miR-containing EVs can apparently be taken up by endometrial cells, but how should this be seen in a pathophysiologic manner? That miR-34c-5p, produced by endometrial cells, can enter the same cells by which it was produced? Or does miR-34c-5p remain internally within the endometrial cells after synthesis and negatively influence GAS1 within the same cell?
Finally, it is not completely clear why miR-210 and miR-582-5p were neglected for follow-up studies, since these were much more specifically upregulated at d.4 (WOI) compared to d.5 (post-implantation), whereas for miR-34c-5p this seemed to be the opposite.
Author Response
Dear Reviewer:
Thank you very much for your careful review and constructive suggestions with regard to our manuscript. Those comments are very helpful for us to revise and improve our paper. We have studied comments carefully and have made correction which we hope meet with approval.
Responses to the reviewers' comments as follows:
All responses were based on the revised manuscript.
Reviewer’s comments
Tan and colleagues have investigated the endometrium and it products in pregnant mice at pre-implantation, implantation, and post-implantation phase. The number of intraluminal vesicles in the endometrium is increased during implantation. They obtained extracellular vesicles (EVs) from the endometrium and showed that these are full of microRNAs, including many that previously have been related to uterine receptivity. Notably, levels of miR-34c-5p in the EVs are increased during the window of implantation (d.4). Analysis of information from miR prediction databases and previous research of miR/mRNA expression of endometrial tissue (normal and RIF) showed that: 1) miR-34c-5p negatively influences GAS1 expression; and 2) implantation failure in humans is related to decreased GAS1 expression and increased miR-34c-5p expression. In their own mouse model the investigators demonstrate increased GAS1 and decreased miR-34c-5p during the window of implantation. Finally, they show that injection of miR-34c-5p agomirs into the uterine horns at d.3 of pregnancy leads to decreased GAS1 expression and a reduced number of implanted embryo (at d.7), confirming that indeed this gene plays a role in embryo implantation.
This is a nice study, incorporating both various practical techniques as well as database mining and mechanistic studies, to verify the hypothesis that microRNAs (in this case 34c-5p) is essential in determining success or failure of embryo implantation.
Point 1: I do have doubts about the title and some of the main conclusions: what is now the evidence that it is the EVs that indicate successful embryo implantation? Previous research, on which data mining was performed by the authors, has been derived from endometrial tissues and not from vesicles. Furthermore, in their own mouse model (Fig. 6), the miR-34c-5p expression was determined in endometrial tissues. In other words, although Fig. 4 shows the expression profiles of miRs in EVs, it is not clear how much of the tissue (cytoplasm) itself contributes to the miR pool and how much is contributed by the EVs. I understand that there is need for biomarkers, possibly miR-containing EVs, of implantation failure, but currently the formulation of the title and some of the statements / conclusions throughout the manuscript should be adjusted (e.g. to something like “miR-34c-5p is enriched in EVs and decreases the success of embryo implantation”).
Response 1: Thank you for your suggestion. We analyzed the recurrent implantation failure dataset and screened genes associated with successful implantation. We found that miR-34c-5p targets GAS1 and has important regulatory effects on embryo implantation. The expression of miR-34c-5p in the endometrium during implantation was extremely low, but the miR-34c-5p content in sEVs was significantly increased at WOI, indicating that miR-34c-5p was packaged in sEVs and secreted into Uterine cavity. Therefore, miR-34c-5p in sEVs is contributed by sEVs, not tissues. This indicates that sEVs miR-34c-5p has potential as a marker.
Due to your suggestion, I found the deficiencies in my current work. We do make inappropriate statements about some results and conclusions, and we have made adjustments in the “Results”and “discussion” sections of revised manuscript.
Point2: In the light of the previous comment, Fig. 3e is not completely clear to me: the miR-containing EVs can apparently be taken up by endometrial cells, but how should this be seen in a pathophysiologic manner? That miR-34c-5p, produced by endometrial cells, can enter the same cells by which it was produced? Or does miR-34c-5p remain internally within the endometrial cells after synthesis and negatively influence GAS1 within the same cell?
Response 2: Thank you for pointing this out.
① Previous studies have shown that extracellular vesicles secreted by parental cells can affect their own physiological characteristics, and that cells have selective absorption of sEVs ( LeBleu, R.K.a.V.S., The biology, function, and biomedical applications of exosomes. Science, 2020. 367(6478)). Uterus secretes different sEVs miRNAs at different stages of early pregnancy (https://doi.org/10.1095/biolreprod.115.134890). Some of these sEVs are present in the uterine fluid and absorbed by the embryo, and some are absorbed by the uterus to regulate the physiological state of the uterus and prepare for embryo attachment. The results in Figure 3e are to investigate whether exosomes of uterine origin can be absorbed by themselves.
②In our study, we found that miR-34c-5p was hardly expressed during implantation, but was highly enriched in sEVs secreted from the endometrium. GAS1 was significantly high expressed during implantation. Therefore, sEVs miR-34c-5p may not enter the same cells by which it was produced. During non-implantation, miR-34c-5p is synthesized in endometrial cells and regulates GAS1 expression. The regulatory mechanism of miR-34c-5p during pregnancy requires further scientific research.
Point3: Finally, it is not completely clear why miR-210 and miR-582-5p were neglected for follow-up studies, since these were much more specifically upregulated at d.4 (WOI) compared to d.5 (post-implantation), whereas for miR-34c-5p this seemed to be the opposite.
Response 3: Thank you for pointing this out. We detected the expression of miR-210 and miR-582-5p in the endometrial tissues and sEVs, as shown below (the tissue expression data is not shown in the manuscript). We found that although the expression of miR-210 in the endometrial tissue was significantly decreased during the window of implantation (WOI), the expression in sEVs could not be distinguished significantly between WOI and post-implantation, so it was not suitable as a marker. In addition, we performed an agomir injection experiment to examine the effect of miR-210 in embryo implantation, however, the data were unstable (data not shown).
Regarding miR-582-5p, we found that its expression was not significant in endometrial tissues during WOI. Most importantly, after detecting the expression of these miRNAs, we learned through database analysis that GAS1 was significantly related to repeated implantation failures, and miR-34c-5p might target GAS1. This shows that miR-34c-5p has a regulatory effect on embryo implantation, so we chose miR-34c-5p.
According to our results, miR-34c-5p is almost not expressed in the endometrial tissues during the period of implantation, but it is highly expressed in sEVs and persists after implantation. We speculate that miR-34c-5p is excreted by the uterus to support implantation. Compared to the expression during non-implantation, miR-34-5p in sEVs has potential as a marker to indicate successful implantation. This result is not inconsistent with previous sEVs uptake experiments, and endometrial uptake of sEVs is also selective. The specific biological mechanism is worthy of our further investigation.
Special thanks to you for your good comments.

Reviewer 3 Report
The objective of the present study was to examine the morphology of mouse endometrium and isolate and characterize the small Extracellular vesicles (sEVs ) derived from pre-implantation, implantation and post-implantation period. Furthermore, identification of sEVs miRNAs derived from endometrium during early pregnancy could be proposed as novel marker for evaluation of endometrial physiology and also could provide new insights for clinical therapy and monitor of infertility.
The manuscript is well written with clear experimental design and results presented and discussed adequately based on the current literature of EVs in reproduction supporting the final conclusion. Using the mice models the authors identified miRNAs in key stages of pre- and post-implantation which may play an important role as markers of endometrial receptivity and pregnancy establishment. The current study provides new knowledge in the area of EVs and their target miRNAs in reproduction using the mouse model, which may be applied in future studies for humans.
Minor comments
Page 1-Line 16: sEVs = Small Extracellular Vesicles
Page 2 - Lines 87 & 91: “sEV” = ??
Page 3 - Line 110, 111, 112, 113 ….: sEVs
Page 3 - Line 121: sEV, sEVs were placed….
“Inconsistence” and confusion between sEV and sets - Please verify and correct were appropriate through the manuscript.
Page 6 - Line 226 - glandularand epithelium epithelium of endometrium (Figure 2).
English need to be corrected by a native speaker.
A few examples:
- Current studies consider that sEVs originated from early endosome and are packaged in the multivesicular bodies (MVBs) [28]. In addition, during the pregnancy, a majority
- In this study, A comprehensive and sequential study of sEVs derived from endometrium during…
- …….released when MVBs fused with plasm membrane [47].
- Previous studies had isolated exosomes/EVs from UF [17], but there lack the study containing whole stages of early pregnancy from early to late secretory.
Author Response
Dear Reviewer:
Thank you for your comments concerning our manuscript. Those comments are all valuable and very helpful for revising and improving our paper, as well as the important guiding significance to our researches. We have studied comments carefully and have made correction which we hope meet with approval.
Responses to the reviewers' comments as follows:
All responses were based on the revised manuscript.
Reviewer’s comments
The objective of the present study was to examine the morphology of mouse endometrium and isolate and characterize the small Extracellular vesicles (sEVs ) derived from pre-implantation, implantation and post-implantation period. Furthermore, identification of sEVs miRNAs derived from endometrium during early pregnancy could be proposed as novel marker for evaluation of endometrial physiology and also could provide new insights for clinical therapy and monitor of infertility.
The manuscript is well written with clear experimental design and results presented and discussed adequately based on the current literature of EVs in reproduction supporting the final conclusion. Using the mice models the authors identified miRNAs in key stages of pre- and post-implantation which may play an important role as markers of endometrial receptivity and pregnancy establishment. The current study provides new knowledge in the area of EVs and their target miRNAs in reproduction using the mouse model, which may be applied in future studies for humans.
Minor comments
Comments 1:
Page 1-Line 16: sEVs = Small Extracellular Vesicles
Page 2 - Lines 87 & 91: “sEV” = ??
Page 3 - Line 110, 111, 112, 113 ….: sEVs
Page 3 - Line 121: sEV, sEVs were placed….
“Inconsistence” and confusion between sEV and sets - Please verify and correct were appropriate through the manuscript.
Response 1: Thank you for your suggestion. We have checked and corrected this problem. “Extracellular vesicles” indicate all types of vesicles. In the revised manuscript, “Extracellular vesicles” was abbreviated as “EVs”.
While “small extracellular vesicles” indicate vesicles with the diameter range from 40to200nm, all “small extracellular vesicles” in the revised manuscript was abbreviated as “sEVs”.
Comments 2:
Page 6 - Line 226 - glandularand epithelium epithelium of endometrium (Figure 2).
English need to be corrected by a native speaker.
Response2: Thank you for your suggestion. We have corrected this sentence. The repeated word “epithelium” has been deleted.
A few examples:
Response: Thank you for your suggestion about the errors of the writing. We have checked those mistakes in manuscript. To further improve the writing quality, we turned to a professional English editing service to modify the sentences, words, and grammar in the paper to ensure its accuracy.
- Current studies consider that sEVs originated from early endosome and arepackaged in the multivesicular bodies (MVBs) [28]. In addition, during the pregnancy, a majority
Response: Thanks for pointing out the mistake. We have revised this sentence. The corrected sentence as follow: “Current studies consider that the sEVs originate from the early endosome and package in the multivesicular bodies (MVBs) [28]. Additionally, during pregnancy…”
- In this study, A comprehensive and sequential studyof sEVs derived from endometrium during…
Response: Thank you for your suggestion. We have corrected this sentence. “In this study” was deleted.
- …….released when MVBs fused with plasm membrane [47].
Response: Thanks for pointing out the mistake. We have corrected ‘plasm membrane’ into ‘Plasma membrane’ in the revised manuscript.
- Previous studies had isolated exosomes/EVs from UF [17], but there lack the study containing whole stages of early pregnancy from early to late secretory.
- Response: Thank you for your suggestion. We have corrected this sentence. The corrected sentence as follow: “In previous studies, exosomes/EVs have been isolated from the UF, but lacking the study containing whole stages of early pregnancy from early to late secretory.”
Once again, thank you very much for your comments and suggestions.
Round 2
Reviewer 1 Report
In this re-submission, the authors have addressed most of the concerns raised in the initial manuscript. I would recommend for publication is the following issues are addressed.
However, I am still not convinced by figure 2. As requested the authors have included a negative control, however the immunostaining appears along the luminal epithelium, just like stained slices. By my eyes, there is no difference. I also see no staining in glands. They have suggested that this is due to brightness/contrast of the image. Can the authors confirm that images has identical settings, and will the published images show clearer staining.. Could the authors downplay their statement that CD63 is expressed in luminal epithelium.
The authors also describe a 'positive (immune cells) control' in the legend but it is not clear which image this is. Could the authors ensure labelling is obvious and appropriate.
Many thanks
Paul
